# Chemical and Antioxidant Characteristics of Skin-Derived Collagen Obtained by Acid-Enzymatic Hydrolysis of Bigeye Tuna (*Thunnus obesus*)

**DOI:** 10.3390/md19040222

**Published:** 2021-04-16

**Authors:** Liza Devita, Mala Nurilmala, Hanifah Nuryani Lioe, Maggy T. Suhartono

**Affiliations:** 1Department of Food Science and Technology, Faculty of Agricultural Engineering and Technology, Bogor Agricultural University, Bogor 16680, Indonesia; liza.devita1981@gmail.com (L.D.); hanifahlioe@apps.ipb.ac.id (H.N.L.); 2The Ministry of Agriculture Republic Indonesia, Jakarta 12550, Indonesia; 3Department of Aquatic Product Technology, Faculty of Fisheries and Marine Sciences, Bogor Agricultural University, Bogor 16680, Indonesia; mnurilmala@apps.ipb.ac.id

**Keywords:** *Thunnus obesus*, by-products, collagen extraction, SDS-PAGE, infrared, free radical, enzyme

## Abstract

The utilization of bigeye tuna skin as a source of collagen has been increasing the value of these skins. In this study, the quality of the skin was studied first. The skin after 14 h freeze-drying showed a high protein level (65.42% ± 0.06%, db), no histamine and a lack of heavy metals. The collagens were extracted through acid and acid-enzymatic methods. The enzymes used were bromelain, papain, pepsin, and trypsin. The two highest-yield collagens were pepsin-soluble collagen (PSC) and bromelain-soluble collagen (BSC). Both were type I collagen, based on SDS-PAGE and FTIR analysis. They dissolved very well in dimethyl sulfoxide and distilled water. The pH ranges were 4.60–4.70 and 4.30–4.40 for PSC and BSC, respectively. PSC and BSC were free from As, Cd, Co, Cr, Cu, and Pb. They showed antioxidant activities, as determined by the DPPH method and the reducing power method. In conclusion, bigeye tuna skin shows good potential as an alternative source of mammalian collagen. Although further work is still required, PSC and BSC showed the potential to be further used as antioxidant compounds in food applications. Other biological tests of these collagens might also lead to other health applications.

## 1. Introduction

Collagen is known as the main protein of vertebrate tissue. There are various types of collagen, with at least 29 types of collagen playing different roles in tissue [1,2]. Each type of collagen has a distinctive amino acid sequence and molecular structure. Collagen type I, characterized by two identical α_1_ chains and one α_2_ chain in the molecular form of [α_1_(I)]_2_α_2_(I), is the most common collagen found in connective tissue, a main component of tendons, ligaments, and bones [3,4]. Collagen type II is collagen present in cartilage tissue. Collagen type III strengthens the walls of hollow structures, which are highly dependent on age. Type IV forms the basal lamina of the epithelia. Other types of collagen are present in very low amounts and are mostly organ-specific [5,6].

Collagen has been applied in the fields of food, cosmetics, and pharmaceuticals [7]. Commercial collagen generally comes from pigs, cows, and poultry, which have drawbacks such as the risk of disease transmission from animals to humans and the presence of religious restrictions on the use of these animals [8,9,10].

Efforts to find alternative sources of collagen beyond pigs, cattle, and poultry have been the focus of much recent research. Nowadays, research on fish-based collagen, in particular from the by-products of the fishery industry, such as skin, scales, and notochord, is much preferred [8,9,11,12,13,14,15,16]. In addition to the quest for alternative sources of collagen, the use of fishery industry by-products will provide added value to the by-products, and will aid in efforts to reduce the cost of managing the environmental impacts that may arise from the usually unused by-products.

Some studies on collagen from fish skin have been reported. These studies examined the fan-bellied leatherjacket (*Monacanthus chinensis*) [17], *Gibelion catla* and *Cirrhinus cirrhosus* [18], brownstripe red snapper (*Lutjanus vitta*) [19], bigeye snapper (*Priacanthus macracanthus*) [20], sole fish [21], parang-parang (*Chirocentrus dorab*) [22], Atlantic Codfish (*Gadus morhua*) and Atlantic Salmon (*Salmo salar*) [23], unicorn leatherjacket (*Aluterus monoceros*) [24], Nile perch (*Lates niloticus*) [25], tilapia (*Oreochromis niloticus*) [26], Northern Snakehead (*Channa argus*) [27], puffer fish (*Lagocephalus inermis*) [28], brown backed toadfish (*Lagocephalus gloveri*) [29], Amur sturgeon (*Acipenser schrenckii)* [30], bester sturgeon [31], Ocellate puffer fish (*Takifugu rubripes*) [32], and channel catfish (*Ictalurus punctatus*) [33].

The characteristics of collagen depend on the material and the extraction method used. The dilute acetic acid method with pepsin is one of the methods used in collagen extraction. The yield of collagen produced using this method is generally greater than the collagen yield using only dilute acetic acid (without pepsin) [11,18,34,35,36]. However, pepsin is also commonly derived from pigs. Therefore, efforts to obtain collagen that can be accepted by all social groups offer an interesting challenge for researchers.

Tuna is one of the most important marine fish in world trade. Indonesia is one of the world’s largest tuna-producing countries. The economic value of the trade in Indonesian tuna fishery products is very large and is an opportunity that can continue to be exploited. The export volume of Indonesian tuna reached 198,131 tons, with a value of 659.99 million USD, in 2017 (https://kkp.go.id/artikel/4409-pesona-tuna-sebagai-penggerak-bisnis-perikanan-indonesia (accessed on 12 August 2019)). Tuna is widely consumed to produce various products such as sashimi and canned food, in which only a certain part of the tuna (flesh/meat) is used in the product. This condition causes many tuna fishery industry by-products to be wasted, which has an impact on the environment. One of the tuna fishery industry by-products is tuna skin.

The study of skin collagen extraction from bigeye tuna, one species of tuna, using a different kind of protease is still limited. To date, to our knowledge, the standard methods for producing collagen from bigeye tuna are extraction with acetic acid containing pepsin [36,37] and extraction using isoelectric precipitation [37]. Since the addition of enzymes helps in the hydrolysis of some structural protein matrices that might still cover the collagen, this study focused on collagen extraction from the skin of bigeye tuna using acetic acid with some proteases (pepsin and others) and also studied the chemical characteristics of the extracted collagen.

The objectives of this study were to determine the quality of the skin as a collagen source through analysis of the proximate composition, histamine, and heavy metal contents of the lyophilized fish skin, as well as the ratio of lyophilized skin to wet skin; to extract collagens from the lyophilized skin of bigeye tuna using acetic acid with and without proteases (bromelain, papain, pepsin, and trypsin); and to study the chemical properties and antioxidant activity of the highest-yield collagen.

## 2. Results and Discussion

### 2.1. Quality of Fish Skin as a Source of Collagen

The quality of the skin of bigeye tuna was explored before collagen extraction. The skins were studied in terms of the proximate composition, as well as the content of histamine and heavy metals that might be present on the skins. Lyophilization was conducted for 10 and 14 h and the ratio of the lyophilized skin to wet skin was calculated, with the aim of finding out how much the conversion affected the ratio of wet skin to lyophilized skin, which would be further used for collagen extraction.

The proximate compositions of the lyophilized skins are shown in Table 1. Protein is the main component of the skin, followed by crude fat, carbohydrate, and ash. The ash, protein, and carbohydrate content increased with increasing lyophilization time, but the fat content decreased with increasing lyophilization time.

The lyophilized skin after 14 h treatment provided a protein content that was higher than the 10 h lyophilized skin, which was 65.42% of the total skin content. Thus, the 14 h lyophilized skin of bigeye tuna was used in our subsequent studies.

Histamine and heavy metal content are shown in Table 2. Histamine and heavy metals analysis showed that the skins were free from histamine, As, Cd, Co, and Pb, but Cr and Cu were detected at tolerable limits in the skins.

Thus, the lyophilized skin of bigeye tuna in our study showed a high content of protein, and was free from the dangers of heavy metals, and no histamine content was detected. For 14 h lyophilization time, a yield of 50.06% ± 0.46% lyophilized skin could be obtained from the certain weights of wet skin.

### 2.2. Extraction and Yield of Collagen

Collagen was isolated from the lyophilized skin of bigeye tuna (*Thunnus obesus*) in two stages. The first stage consisted of pre-treatment using 0.1 M NaOH (in distilled water), followed by the use of 10% butyl alcohol (in distilled water), with the aim of removing the non-collagen components, such as non-collagen proteins, fats, minerals, pigments, and odors. In the second stage, the treated skins were soaked in 0.5 M acetic acid with and without protease (bromelain, papain, pepsin, and trypsin), followed by precipitation by NaCl, separation and collection of the precipitating material using a refrigerated centrifuge, then samples were dialyzed and lyophilized. The collagen produced was designated as acid-soluble collagen, bromelain-soluble collagen, papain-soluble collagen, pepsin-soluble collagen, and trypsin-soluble collagen, assigned to collagens obtained with extraction using only 0.5 M acetic acid, and 0.5 M acetic acid containing bromelain, papain, pepsin, and trypsin, respectively.

The collagen yields can be seen in Table 3. Collagen extraction using 0.5 M acetic acid (dilute acetic acid) gave the lowest yield compared to the collagen extraction using 0.5 M acetic acid containing protease enzymes. The use of limited proteolysis by bromelain or pepsin in 0.5 M acetic acid was able to produce high-yield collagen. However, limited proteolysis by adding papain or trypsin to 0.5 M acetic acid was only able to produce low-yield collagen (but its yield was still higher than collagen extracted using only 0.5 M acetic acid). These results show that the lyophilized skin of bigeye tuna was not soluble enough with an extraction process that used only 0.5 M acetic acid, but was sufficient and very soluble with an extraction process that used 0.5 M acetic acid containing papain, trypsin, bromelain, and pepsin.

As stated above, the acid-soluble collagen and pepsin-soluble collagen were the collagens with the lowest and highest yields, respectively. This can be explained as follows. Collagens have non-helical parts on the telopeptide region at both ends of the N terminal and the C terminal, which is a cross-linked structure [36,37,38]. This cross-linked structure limits its solubility in 0.5 M acetic acid without the presence of pepsin. In this case, pepsin specifically cleaved the telopeptide, resulting in high collagen yields. The structure of pepsin-soluble collagen is shown in Figure 1 (structure observed with the naked eye).

These results are in agreement with previous studies of collagen isolation from fishery industry by-products, including skin, which also reported low solubility in 0.5 M acetic acid solutions and high solubility in pepsin-containing acetic acid, as obtained from the same species, bigeye tuna (*Thunnus obesus*) [36,39], and other species: ocellate puffer fish (*Takifugu rubripes*) [32], channel catfish (*Ictalurus punctatus*) [33], giant croaker (*Argyrosomus japonicus*) [38], cuttlefish (*Sepia lycidas*) [40], paper nautilus (*Argonauta argo, Linnaeus*) [41], minke whale (*Balaenoptera acutorostrata*) [42], *Sepia pharaonis* (Ehrenberg, 1831) [1], and *Sepiella inermis* (Orbigny, 1848) [43]. However, it was not in agreement with studies of northern snakehead (*Channa argus*) [27].

Reports on collagen extraction with bromelain, papain, and trypsin are still limited. Different proteases have different activities. They are specific in cleaving certain peptide bonds. Bromelain works efficiently on synthetic substrates containing Arg-Arg [44,45]. Papain, with wide specificity, catalyzes peptide bond hydrolysis. This enzyme binds to substrates that have a large hydrophobic side chain at the P2 position and will not accept valyl residues at the P1 position. Papain also shows an esterase and an amidase activity. Under favorable conditions, papain catalyzes the formation of peptide bonds [46]. Pepsin shows preferential cleavage of aromatic residues in either position of the peptide bond [47]. Trypsin hydrolyzes the peptide and ester bonds formed by the carboxyl group of the base amino acid, including Arg and Lys [47,48]. Overall, all of these proteases affected collagen yields.

Interestingly, we found that 0.5 M acetic acid containing bromelain was able to produce a high collagen yield, similar to that of PSC. Bromelain is rarely used in collagen isolation. This becomes interesting when associated with halal issues because, unlike animal pepsin, bromelain is a protease derived from plants. Since bromelain-soluble collagen and pepsin-soluble collagen were the two collagens with the highest yields, in our next study, we only used these two collagens (collagen samples).

### 2.3. Chemical Properties of Collagen

#### 2.3.1. Heavy Metal Content

It should be noted that in the collagen isolation process a pre-treatment process was carried out, of which the function was to remove the non-collagen substances, including in this case, removing the presence of metals. Based on the Cr and Cu tests on the collagen samples extracted from the skin of bigeye tuna (the results are shown in Table 4), Cr and Cu were not detected. Thus, it can be ascertained that collagen does not contain As, Cd, Co, Cr, Cu, and Pb (As, Cd, Co, and Pb were not detected in the skin of bigeye tuna, and were certainly also not present in collagen samples). We conducted analysis to ensure that the generated collagen was fully free of heavy metals.

#### 2.3.2. Solubility

Collagen solubility is the ability of collagen to dissolve in solvents. In the present study, collagens’ solubility in several solvents were studied in dimethyl sulfoxide, distilled water, ethanol, and methanol. Solvents used in these tests were selected based on several considerations: they were non-toxic or had low toxicity, had a high ability to solubilize compounds in general, and were cheap and easy to obtain and thus would be useful for further industrial applications of collagen.

The solubilities of the lyophilized collagen samples in dimethyl sulfoxide, distilled water, ethanol, and methanol are shown in Table 5. The collagen samples were able to dissolve well in dimethyl sulfoxide and distilled water, but they were only slightly soluble in ethanol and methanol.

Solubility testing of skin collagen from another tuna species, yellowfin tuna (*Thunnus albacares*), has been reported previously [49]. Those researchers reported that collagen solubility in distilled water was within the range of 78.67%–95.27%. Our results were in line with this earlier report.

The solubility level of a solute in solvents depends on the chemical and physical properties of the solute and the solvent. Thus, the solubility level of collagen samples in these solvents could be explained by the principle of “likes and dislikes”, which are determined by the suitability of the properties between the solvents and the collagen samples. Notably, the collagen samples were dissolved in all of these solvents, which are polar compounds.

#### 2.3.3. pH

The pHs (*n* = 3) of pepsin-soluble collagen and bromelain-soluble collagen were in the range of 4.60–4.70 and 4.30–4.40, respectively. These results are slightly different from the previous study [49] with another species of tuna, which showed that the pHs of yellowfin tuna skin-derived collagen were in the range of 5.90–5.96. This might be due to the difference in the collagen isolation process and the skin that was used. Overall, however, the pH value was still in the acidic range (pH < 7).

#### 2.3.4. Protein Profile Determined by SDS-PAGE

The results of the SDS-PAGE analysis of collagen samples from bigeye tuna skin are shown in Figure 2. The electrophoretic patterns revealed that the collagen samples had at least two different α chains, namely α_1_ and α_2_. The α_1_ and α_2_ protein bands for our collagen samples displayed two clear bands (123 and 115 kDa) in lane 2 and two clear bands (120 and 110 kDa) in lane 3, for pepsin-soluble collagen and bromelain-soluble collagen, respectively.

These results were consistent with other studies analyzing fish skin collagens, which also showed at least α_1_ and α_2_ bands, respectively, at 120 and 110 kDa [17], 120 and 100 kDa [30], and α chains in the range of 116–118 kDa [21]. Our result was also in agreement with the pattern revealed by the collagen protein profiles of the same species (*Thunnus obesus*) but from a different organ (bone) in a previous study [40]. Their collagen (pepsin-soluble collagen) showed α_1_ (130 kDa) and α_2_ (115 kDa) bands, as well as other protein bands of β and γ. In accordance with this, the β and γ protein bands were also obtained in our collagen study, although they appeared less clearly. The enzymes used (pepsin and bromelain) may hydrolyze the cross-link in the β-chain. The α_3_ may shift to the same position as α_1_, such that the α_3_ chain could not be viewed as clearly. The electrophoretic pattern showed that pepsin-soluble collagen and bromelain-soluble collagen were type I collagens.

#### 2.3.5. Functional Group Analysis Using FTIR

Infrared (IR) spectroscopy is used in many qualitative and quantitative studies [50]. In IR spectroscopy, IR absorption happens only when those vibrations alter the electric dipole moment [51]. Motions of stretching and bending (scissoring, rolling, spinning, and wagging) are the most important motions that create a change in the dipole moment in a polyatomic molecule [52].

Fourier transform infrared (FTIR) spectroscopy, a particular method of infrared spectroscopy, provides information at the molecular level that allows for the analysis of functional groups, bonding types, and conformations. The FTIR spectroscopy instrument is called an FTIR spectrometer. This technique is simple, non-destructive, reproducible, and requires a small sample amount [53].

FTIR analysis at 400–4000 cm^−1^ [50] was used to confirm samples as collagens. Collagens displayed characteristic bands (amide A, B, I, II, and III) in the FTIR analysis. Figure 3 displays the FTIR spectrum of the collagen samples. The spectrum showed a region of group frequencies and a fingerprint area [51]. The interpretation of the FTIR spectrum is presented in Table 6. Based on this analysis, there were considerable similarities between the FTIR spectra of the two collagen samples, with slight differences seen in the fingerprint area. This shows that the two collagen samples were similar, but slightly different for the specific functional groups present in the fingerprint area (1500–400 cm^−1^) [54], especially in the area ranging from 400–750 cm^−1^.

#### 2.3.6. Protein Content of Collagen

The protein contents of the collagen samples were analyzed using the Lowry method. The results were 6.58 ± 0.01 and 16.37 ± 0.01 mg/mL for pepsin-soluble collagen and bromelain-soluble collagen, respectively. All values are given as mean ± standard deviation, *n* = 2. Values of protein content showed significant differences between PSC and BSC, determined by one-way ANOVA (*p* < 0.05).

### 2.4. Antioxidant Activity of Collagen

The antioxidant activities of the collagen samples were studied with the DPPH radical scavenging test (2,2-diphenyl-1-picrylhydrazyl) and the reduction power test. Thus, the antioxidant activities of collagen samples were studied using two different methods. The base principles of the two methods were as follows. (1) DPPH radical scavenging test: DPPH provided maximum absorbance at a wavelength of 517 nm. The presence of antioxidant properties in collagen samples was indicated by a decrease in absorbance or a decrease in the intensity of the purple color. (2) Reduction power test: There were direct correlations between the antioxidant activities of the collagen samples and their ability to reduce iron (III) to iron (II). The formation of the iron (II) complex was monitored by measuring the formation of Prussian Perl blue at 700 nm through absorbance values [59,60]. Thus, in this reducing power test, the antioxidant activities of collagen samples were proportional to the absorbance values.

The antioxidant activities of the collagen samples are summarized in Table 7 and Table 8. Table 7 shows the collagens’ antioxidant activities as determined using the DPPH radical scavenging method, and Table 8 shows the collagens’ antioxidant activities as determined using the reducing power method. Our findings showed that both collagen samples, bromelain-soluble collagen and pepsin-soluble collagen, had antioxidant activities, as determined either by DPPH radical scavenging test or by reducing power test. The antioxidant activity of pepsin-soluble collagen was higher than the antioxidant activity of bromelain-soluble collagen as determined by the DPPH radical scavenging test. However, the antioxidant activity of the two collagen samples did not show a significant difference according to the reducing power method. In general, the antioxidant activities of the two collagen samples were lower than the antioxidant activities of ascorbic acids as determined by both methods. Even though our work is presently limited to food applications of collagen, additional biological testing of our collagen, such as its molecular effect on keratinocystes and other cell types, might be useful in order to pursue health applications, as collagen is known to play a very important role in wound healing and skin cell renewal.

The antioxidant activities of collagens have also been studied by previous researchers. ASC and PSC (0.25–10 mg/mL) obtained from *Argyrosomus japonicus* swim bladders were able to scavenge DPPH, hydroxyl, superoxide anion, and ABTS radicals [38]. Our findings were consistent with the results of their studies, as our collagen samples also showed antioxidant activities, although at relatively lower values. Nevertheless, these data have encouraged us to pursue further studies, adding other protease enzymes in order to hydrolyze PSC and BSC to produce smaller peptides, and to further analyze their antioxidant activity. This will be the focus of our next experiments.

The antioxidant activity of the bigeye tuna skin collagen could be explained as follows. Based on the literature, collagen from the skin of bigeye tuna is dominated by several amino acids, namely glycine, proline, glutamic acid, alanine, and hydroxyproline. This collagen also contains other amino acid residues, such as arginine, aspartic acid, cysteine, histidine, hydroxylysine, isoleucine, leucine, lysine, methionine, phenylalanine, serine, threonine, tyrosine, valine, and imino acids [36]. Several amino acids have been reported to be influential in the activity of free radical scavenging, including Trp, Tyr, Phe, Lys, and Arg [48]. Thus, the DPPH radical scavenging activity shown by our collagen samples was possibly due to the presence of the amino acids arginine, lysine, phenylalanine, and tyrosine. The richness of these potential amino acids in our collagen is implied in the results of our FTIR analysis.

## 3. Materials and Methods

### 3.1. Materials

The skin of bigeye tuna (*Thunnus obesus*) was obtained from PT Maluku Prima Makmur, Ambon, Indonesia. Other materials were acetic acid (glacial) 100%, sodium hydroxide, 1-butanol, papain (30,000 USP-U/mg; EC 3.4.22.2), pepsin (700 FIP-U/g, 0.7 Ph Eur-E/mg; EC 3.4.23.1), sodium chloride GR for analysis, Uvasol^®^ potassium bromide for IR spectroscopy, L(+)-ascorbic acid, ethanol absolute for analysis, dimethyl sulfoxide for analysis, methanol gradient grade for liquid chromatography, acrylamide, N,N,N’,N’-tetramethyl ethylenediamine (Temed), Coomassie^®^ brilliant blue R 250 for electrophoresis (Trademark of Imperial Chemical Industries PLC), nitric acid (65%), and 2-mercaptoethanol, purchased from Merck (Darmstadt, Germany); as well as 2,2-diphenyl-1-picrylhydrazil, glycerol 99.5%, ammonium persulfate, bromophenol blue, phthaldialdehyde, sodium dodecyl sulfate, tris-HCl, bovine serum albumin, glycine, trichloroacetic acid, bromelain from pineapple stem (≥3 units/mg protein; EC 3.4.22.32), and yttrium, purchased from Sigma-Aldrich Pte Ltd. (Singapore); trypsin 1:250 from bovine pancreas (lyophil. Mr 24,000.00), purchased from SERVA Electrophoresis GmbH (Heidelberg, Germany); Spectra™ Multicolor Broad Range Protein Ladder from Thermo Scientific (New York, NY, USA); and distilled water, purchased from PT IKA Pharmindo Putramas (Jakarta, Indonesia).

### 3.2. Preparation of Fish Skin

The collection and preparation of skins were carried out according to the method of [61]. Bigeye tuna skins were transported from Ambon to the research laboratory by plane for 5 h in an icebox, then stored at −28 °C in a freezer. These skins were separated from the flesh, cleaned, and cut into small pieces (0.5 × 0.5 cm^2^). They were then lyophilized using a TFD5503 Bench-Top freeze dryer (ilShinBioBase, Ede, The Netherlands). The lyophilized fish skins were used for proximate analysis and collagen extraction.

### 3.3. Quality Determination of Fish Skin as a Source of Collagen

Measurements of the proximate composition, histamine content and heavy metal content of lyophilized skin, and the ratio of lyophilized skin to wet skin were carried out using the following methods.

Determination of the ratio of lyophilized skin to wet skin was carried out according to the following procedure. Approximately 100 g of wet skin that had been cleaned and cut into pieces was lyophilized using a TFD5503 Bench-Top freeze dryer (ilShinBioBase, Ede, The Netherlands) with a time variation of 10 h and 14 h, then weighed. The ratio of lyophilized skin to wet skin was as follows:Ratio (%) = (weight of lyophilized fish skins)/(weight of wet fish skins) × 100(1)

The lyophilized fish skins were used for proximate analysis. Moisture, protein, ash, and fat contents were determined following the methods of [62]. Moisture content was determined using the gravimetric method by drying the sample at 105 °C ± 2 °C until a constant weight was attained. The Kjeldahl method was used for the determination of crude protein content (conversion factor of 6.25 × N). Crude lipid content was determined using the Soxhlet method. Ash content was determined after combustion for 20 h at 550 °C. Total carbohydrates were determined by subtracting the sum of fat, protein, moisture, and ash contents from 100.

Histamine testing was carried out by referring to the method of [63] with slight modifications. A sample (5 g) was placed in a 25-mL volumetric flask, diluted with 5% of TCA, homogenized, sonicated (15 min), and replaced in a 2-mL Falcon tube and centrifuged at 14,000 rpm for 3 min. Then, the supernatant (1 mL) was diluted to 10 mL with 1 N NaOH (0.4 mL) and distilled water (8.6 mL) in a 10-mL volumetric flask. Finally, this solution was filtered with a PVDF 0.45-μm syringe filter. Derivatization: 200 μL of standard series and sample solutions were placed into vials of 2 mL, with the addition of 900 μL 3-mercaptopropionic acid (MPA), 440 μL o-phthalaldehyde (OPA), and 50 μL ABBA. The mixtures were homogenized and injected immediately into a high-performance liquid chromatography (HPLC) system. The measurement conditions of the HPLC instrument were as follows. Waters Atlantis T3 (HPLC Alliance E 2695 w/FLD 2475 Waters with software empower 3), 150 mm × 4.5 mm, 5 μm column, 0.1 M acetate buffer pH 4.7 ± 0.05 (A) and Acetonitrile (B) (mobile phase), 1.2 mL/min (flow rate), gradient (pump system), 10 μL injection volume, 40 °C (detector), FLD (λ _excitation_ = 330 nm; λ _emission_ = 440 nm).

The formula for the determination of histamine content in the sample, based on the standard calibration curve, which has the line equation y = bx + a, was:(2)$$\mathrm{Histamine}\;\mathrm{content}\;\left(\mathrm{ppm},\frac{\mathrm{mg}}{\mathrm{kg}},\frac{\mathrm{mg}}{L}\right)=\frac{\frac{\mathrm{As}-a}{b}\times V\times \mathrm{df}}{\mathrm{Ws}\;\mathrm{or}\;\mathrm{vs}.}$$

As = sample area, a = intercept, b = slope, df = dilution factor of sample, V = final volume of sample solution (mL), Ws = weight of sample (g), and vs. = volume of sample (mL). The heavy metals content of fish skin was tested using the method of [64]. As, Cd, Co, Cr, Cu, and Pb determinations were performed on an inductively coupled plasma optical emission spectrometry (ICP-OES) Agilent Model 720 with a computer system (software ICP Expert II version 2.0.4). Details of the instrumental operating conditions are given in Table 9. The preparation of the sample was carried out using the following procedure: 0.5–1.0 g of sample was inserted into a vessel, with the addition of 10 mL of concentrated HNO_3_, and placed in a microwave digestion device (ramped up to 150 °C for 10 min and kept at 150 °C for 15 min). The digestion results were replaced into a 50-mL volumetric flask, with the addition of 0.50 mL internal standard yttrium 100 mg/L, diluted with distilled water until marked. The solution was then homogenized, filtered with filter paper, and then measured using the ICP OES system.

The heavy metals content in the sample, based on the standard calibration curve, which has the line equation y = bx + a, was determined using the following formula:(3)$$\mathrm{heavy}\;\mathrm{metals}\;\mathrm{content}\;\left(\mathrm{ppm},\frac{\mathrm{mg}}{\mathrm{kg}},\frac{\mathrm{mg}}{L}\right)=\frac{\frac{\mathrm{As}-a}{b}\times V\times \mathrm{df}}{\mathrm{Ws}\;\mathrm{or}\;\mathrm{vs}.}$$
where As = intensity of sampel, a = intercept of the standard calibration curve, b = slope of the standard calibration curve, df = dilution factor of sample, V = final volume of sample (mL), Ws = weight of sample (g), and vs. = volume of sample (mL).

### 3.4. Extraction and Yield of Collagen

Pre-treatment and collagen extraction were carried out by referring to [61], with slight modifications. Pre-treatment was carried out by soaking the skin in 0.1 M NaOH solution, followed by a 10% (*v*/*v*) butyl alcohol solution. Soaking was carried out at 4 °C for 24 h. The ratio of skin to the solution was 1:10 (*w*/*v*). The NaOH and butyl alcohol solutions were changed every 12 h. The pre-treated skin was then extracted using 0.5 M acetic with and without 0.1% (*w*/*v*) proteases (bromelain, papain, pepsin, and trypsin) at 4 °C for 72 h. The ratio of pre-treated skin to extracting solution was 1:40 (*w*/*v*). The suspension was then centrifuged using a refrigerator centrifuge Hermle Z 32 HK (Benchmark Scientific, Sayrevill, NJ, USA) at 6000× *g* for 30 min at 4 °C. The supernatant was salted-out by adding NaCl to final concentrations of 2.0 M. The precipitate was collected by means of centrifugation at 6000× *g* for 30 min at 4 °C and then dialyzed with cold distilled water using a dialysis membrane, Spectra/Por^®^ Dialysis Membrane MWCO 6–8,000, Regenerated Cellulose (Spectrum Laboratories, Inc., Rancho Dominguez, CA, USA). The wet collagen was then lyophilized using the TFD5503 Bench-Top freeze dryer (ilShinBioBase, Ede, Netherlands). Collagen yields were calculated based on a dry basis:Yields (%) (wb) = (weight of lyophilized collagen)/(weight of lyophilized fish skin) × 100(4)
Yields (%) (db) = (collagen yield of wet basis)/(100 − moisture content lyophilized fish skin) × 100(5)

Collagen with the highest yield, called the collagen sample, was used for further research.

### 3.5. Analysis of Collagen Chemical Properties

#### 3.5.1. Heavy Metal Content Measurement

The heavy metal content of the collagen samples was measured using the inductively coupled plasma optical emission spectrometry (ICP-OES) system [64]. The method for calculating the heavy metal content of collagen samples was the same as that outlined in Section 3.3. The metals measured were Cu and Cr.

#### 3.5.2. Solubility Measurement

Solubility measurements of collagen samples were conducted with several solvents (dimethyl sulfoxide, distilled water, ethanol, and methanol) by referring to [65]. Glass centrifuge tubes were dried at 105 °C and weighed. A number of lyophilized collagen samples (about 40 mg) were dissolved in 2 mL of solvents in the tubes, homogenized for 15 min, sonicated for 15 min, and centrifuged with Hermle Z 32 HK (Benchmark Scientific, Sayreville NJ, USA) for 15 min at 6000× *g* at 25 °C. The supernatants were taken out, and the tubes containing residue were dried in the oven at 105 °C and weighed.

Solubility was determined based on the following equation:Insolubility (%) = (weight of residue/weight of sample) × 100(6)
Solubility (%) = 100 − insolubility (%)(7)

#### 3.5.3. pH

PH measurements of collagen samples were made using a slight modification of the method of [49], i.e., we used wet collagen. Wet collagen samples were melted at room temperature (25 °C). The pH of the collagens was measured using a digital pH meter.

#### 3.5.4. Protein Pattern Analysis by SDS-PAGE

Analysis of protein patterns using sodium dodecyl sulfate polyacrylamide gel electrophoresis (SDS-PAGE) was carried out according to the method if [66], and collagen samples were prepared using the method of [61]. SDS-PAGE was performed using 5% stacking gel and 8% separating gel. Collagen samples were dissolved in 5% SDS to obtain a final concentration of 2 mg/mL. The mixtures were boiled for 5 min in 85 °C and then centrifuged using a Hermle Z 32 HK refrigerator centrifuge (Benchmark Scientific, Sayreville, NJ, USA) at 10,000 rpm for 5 min at 4 °C. The supernatant was taken and dissolved 1:1 with solution B. Solution B contained 125 µL 1 M Tris-HCl pH 6.8, 100 µL glycerol 50%, 200 µL SDS 10%, 50 µL 2-merkaptoetahol, 25 µL 1% bromophenol blue, and 500 µL H_2_O deionized. Then, the samples (15 µL) were loaded into wells of a polyacrylamide gel. Electrophoresis using Hoefer scientific instruments (Hoefer, Inc., San Francisco, CA, USA) was run at 100 volts for about 120 min. After electrophoresis, the gel was stained with 1 g Coomassie brilliant blue R-250 in ethanol 150 mL, acetic acid glacial 50 mL, and miliQ 300 mL for 2 h and then de-staining was carried out with 7.5% (*v*/*v*) of acetic acid and 5% (*v*/*v*) methanol. A Thermo Fisher Scientific Spectra™ Multicolor Broad Range Protein Ladder was used in estimating the molecular weight of protein bands.

#### 3.5.5. Functional Group Analysis Using FTIR Spectrophotometry

Fourier transform infrared (FTIR) analysis of collagen samples was performed by following the method of [24]. Mixtures of KBr and freeze-dried collagen samples with a ratio of 9:1 were placed in DRS cells. The IR spectra (400–4000 cm^−1^) were obtained in 45 scans at a resolution of 2 cm^−1^ and were compared against a background spectrum recorded from the clean empty cell at 25 °C. The data were saved on data files. The FTIR profiles of the collagen samples were recorded using a IRPrestige-21 Shimadzu Fourier Transform Infrared Spectrophotometer (Shimadzu Corporation, Kyoto, Japan).

#### 3.5.6. Protein Content Determination of Collagen Samples

Protein measurement of collagen samples was done using the method of [67]. Absorbance was read at 500 nm with a Jenway 7315 Spectrophotometer (Bibby Scientific Ltd., Staffordshire, UK). Unknown protein concentrations of collagen samples were estimated by plotting on the standard curve in bovine serum albumin.

### 3.6. Antioxidant Activity Tests of Collagen Samples

Antioxidant activities of the collagen samples were investigated by means of the DPPH radical scavenging activity test and the reducing power test.

The antioxidant activity test was performed using the DPPH radical scavenging test based on the method of [68], with slight modifications. Each collagen sample was dissolved in ethanol (1 mg mL^−1^). About 100 µL of the solutions were mixed with 100 µL of DPPH (125 µM in ethanol). The sample mixtures were shaken and incubated at room temperature for 30 min in darkness. The absorbance was measured using a microplate reader (Epoch Microplate Spectrophotometer, BioTek Instruments, Inc., Winooski, VT, USA) at a wavelength of 517 nm. Antioxidant activities were expressed in µmol ascorbic acid equivalent/g protein or mg ascorbic acid equivalent/g protein.

The reducing power test was carried out according to the method of [60] in order to understand the ability of collagen samples to reduce iron (III). Aliquots of 1 mL collagen samples were added to 2.5 mL of 0.2 M phosphate buffer, pH 6.6, and 2.5 mL of 1% (*w*/*v*) potassium ferricyanide. The assay mixtures were homogenized and incubated for 30 min at 50 °C. Then, 2.5 mL of 10% (*w*/*v*) trichloroacetic acids were added and the mixtures were centrifuged at 1650× *g* for 10 min. Lastly, 2.5 mL of the supernatants were mixed with 2.5 mL of distilled water and 0.5 mL of 0.1% (*w*/*v*) ferric chloride solution. Higher absorbance implied a higher reducing power.

### 3.7. Statistical Analysis

Data are presented as the mean ± standard deviation (SD). The data were calculated using IBM SPSS Statistics Version 20 for Windows (SPSS Inc., Chicago, IL, USA) for the Duncan test and Microsoft Excel 2016 (Microsoft Corp, Redmond, WA, USA) for the one-way ANOVA.

## 4. Conclusions

Several proteases (bromelain, papain, pepsin, and trypsin) were successfully applied to collagen extraction from the skin of bigeye tuna (*Thunnus obesus*). Pepsin and bromelain treatment resulted in the highest collagen yields. They were confirmed as type I collagens using SDS PAGE and FTIR analysis. Both collagens had antioxidant activities. The important point of this study is that collagens from tuna bigeye skin can be used as an alternative to mammalian collagen and can be used in food applications. The high yield of bromelain-soluble collagen is an interesting finding, because this extraction uses plant protease, which is related to halal dietary restrictions in Islam. Finally, and most importantly, lyophilized fish skin is highly effective to be used in the study of collagens.

## Figures and Tables

**Figure 1 marinedrugs-19-00222-f001:**
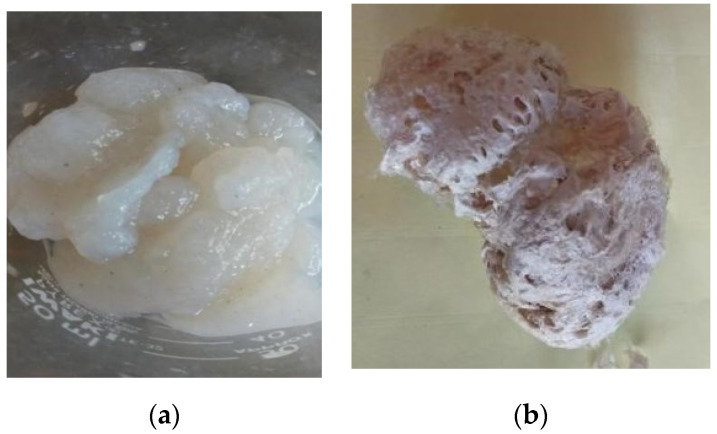
(**a**) Wet and (**b**) lyophilized collagen samples (pepsin-soluble collagen, the extracted collagen with the highest yield).

**Figure 2 marinedrugs-19-00222-f002:**
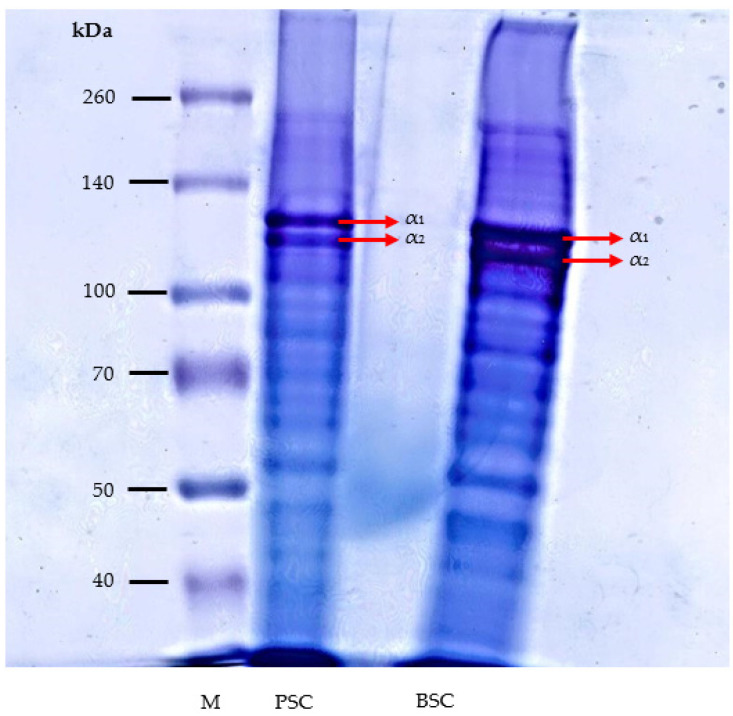
SDS-PAGE patterns of pepsin-soluble collagen and bromelain-soluble collagen from bigeye tuna skin. M, PSC, and BSC are high protein markers, pepsin-soluble collagen, and bromelain-soluble collagen, respectively. Arrows show the collagen bands.

**Figure 3 marinedrugs-19-00222-f003:**
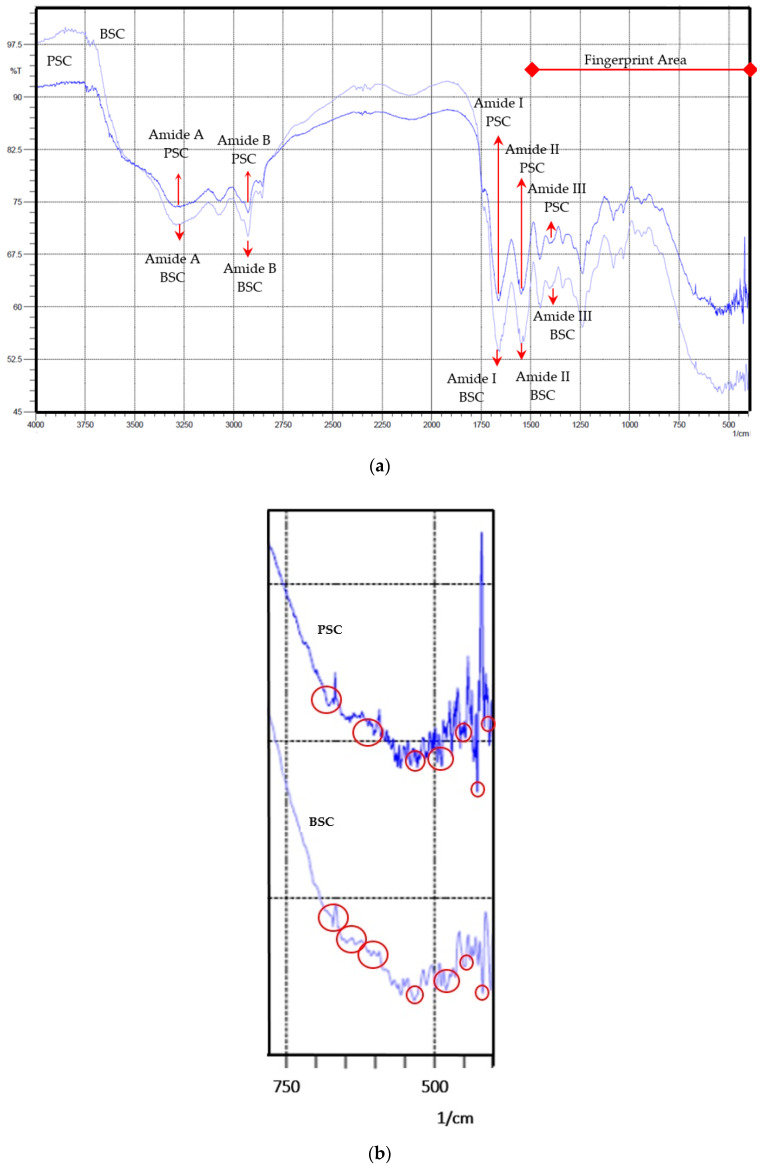
The infrared spectra of collagen samples (**a**) and fingerprint area magnification of the infrared spectra of collagen samples (**b**). The *x*-axis represents the wave number (W) and the *y*-axis represents the percentage of transmittance (%T). PSC and BSC are pepsin-soluble collagen and bromelain-soluble collagen, respectively. The fingerprints distinguishing between PSC and BSC are shown in the circled areas.

**Table 1 marinedrugs-19-00222-t001:** Proximate composition of the lyophilized skins of bigeye tuna.

Fish Skin (db)	Ash (%)	Carbohydrate (%)	Fat (%)	Protein (%)
Lyophilized fish skin 1 (10 h)	1.68 ± 0.37	3.35 ± 0.56	37.02 ± 0.61	57.96 ± 0.32
Lyophilized fish skin 2 (14 h)	1.95 ± 0.28	7.21 ± 0.37	25.42 ± 0.14	65.42 ± 0.06

All values are presented as mean ± standard deviation, *n* = 2. Values of fat, protein, and carbohydrate content showed significant differences between 10 and 14 h, determined by one-way ANOVA (*p* < 0.05).

**Table 2 marinedrugs-19-00222-t002:** Histamine and heavy metal content of the lyophilized skins of bigeye tuna.

Parameter	Results (mg/kg)	Limit of Detection (mg/kg)
As	not detected	0.008
Cd	not detected	0.00011
Co	not detected	0.0008
Cr	0.43 ± 0.02	-
Cu	30.38 ± 0.34	-
Pb	not detected	0.009
Histamine	not detected	1.09

All values are presented as mean ± standard deviation, *n* = 2.

**Table 3 marinedrugs-19-00222-t003:** Yields of extracted collagens from the lyophilized skins of bigeye tuna (dry weight basis).

Extracted Collagens	Yields (db) (%)
Acetate acid soluble collagen	3.05 ± 0.82 ^a^
Bromelain soluble collagen	42.76 ± 4.72 ^c^
Papain soluble collagen	15.20 ± 6.27 ^b^
Pepsin soluble collagen	52.02 ± 0.59 ^c^
Trypsin soluble collagen	13.83 ± 1.95 ^b^

All values are given as mean ± standard deviation, *n* = 2. Values with different letters in the same column indicate significant differences, determined by the Duncan test (*p* < 0.05).

**Table 4 marinedrugs-19-00222-t004:** Heavy metal contents of the collagen samples.

Sample	Parameter	Results (mg/kg)	Limit of Detection (mg/kg)
Bromelain-soluble collagen	Cr	Not detected	0.04
	Cu	Not detected	0.001
Pepsin-soluble collagen	Cr	Not detected	0.04
	Cu	Not detected	0.001

All values are given as mean ± standard deviation, *n* = 2.

**Table 5 marinedrugs-19-00222-t005:** The solubilities of the lyophilized collagen samples in several solvents at 25 °C.

Solvent	Solubility	
	Bromelain-Soluble Collagen	Pepsin-Soluble Collagen
Dimethyl sulfoxide	73.48 ± 1.09 ^d^	76.30 ± 0.41 ^d^
Distilled water	64.30 ± 0.15 ^c^	63.24 ± 0.86 ^c^
Ethanol	48.56 ± 1.03 ^b^	48.42 ± 1.03 ^b^
Methanol	26.85 ± 0.98 ^a^	25.44 ± 2.44 ^a^

All values are given as mean ± standard deviation, *n* = 3. Values with different letters in the same column indicate significant differences, determined by Duncan test (*p* < 0.05).

**Table 6 marinedrugs-19-00222-t006:** Characteristics of functional groups of collagens samples.

Region	Abs. Area (cm^−1^)	Bigeye Tuna Skins			Others Study	
		Peak Wave Number (cm^−1^)		Assign.	Peak Wave Number (cm^−1^)	Assign.
		BSC	PSC			
Amide A	3440–3400 [8,22]	3272	3276	The amide A band is associated with the frequency of stretching N–H. N–H stretching vibration frequency changed from a free N–H stretching vibration frequency (3440–3400 cm^−1^) to a lower frequency in our collagen samples (3272 and 3276 cm^−1^, for BSC and PSC, respectively), which indicated their involvement in the hydrogen bonding. [55,56]	3296 (ASC), 3281 (HWM), and 3271 (SHM) [26]	NH stretch coupled with hydrogen bond
					3425.57 (PSC-IP) [37]	free N–H vibrations as an indication of hydrogen bonds
					3425.58 (ASC) [22], 3440 (ASC Rohu) and 3440 (ASC Catla) [57]	NH stretching
Amide B	2940–2922 [22,24]	2930	2930	The amide B bands were shown at 2930 cm^−1^ for BSC and PSC, indicated CH_2_ asymmetrical stretching.	2924.09(ASC) [22]	CH_2_ asymmetrical stretching
					2938 (ASC), 2939 (HWM), and 2936 (SHM) [26]	Asymmetrical stretch of CH_2_ and ${\mathrm{NH}}_{3}^{+}$
					2930.97 (PSC-IP) [37], 2923 (ASC Rohu) and 2925 (ASC Catla) [57]	CH_2_ asymmetrical stretch
					2858 (ASC Rohu) and 2856 (ASC Catla) [57]	CH_2_ Asymmetrical stretching
Amide I	1690–1625 [22,56]	1665	1660	The amide I bands positions were 1665 and 1660 cm^−1^ (for BSC and PSC, respectively), fitting well the range of 1690–1625 cm^−1^ for general amide I bands position [55,56]^.^	1647.21 (ASC) [22]	C=O stretching
					1631 (ASC), 1629 (HWM), and 1628 (SHM) [26]	C=O stretch/hydrogen bond coupled with ${\mathrm{COO}}^{-}$
					1646.26 (PSC-IP) [37]	C=O stretching vibration on the main polypeptide chain or the hydrogen bond coupled with ${\mathrm{COO}}^{-}$
					1653 (ASC Rohu) and 1643 (ASC Catla) [57]	C=O stretch/hydrogen bond coupled with COO−
Amide II	1600–1550 [58]	1545	1547	The amide II bands were found at 1545 cm^−1^ and 1547 cm^−1^ (BSC and PSC, respectively). When compared to the normal absorption range of the amide II bands’ position (1600–1550 cm^−1^), these positions shifted to a lower frequency, indicating the presence of hydrogen bonds in the collagens [56].	1543.05 (ASC) [22]	NH bend CN stretch
					1544 (ASC), 1536 (HWM), and 1536 (SHM) [26], 1558, 1540 (ASC Rohu) and 1558, 1540 (ASC Catla) [57].	NH bending coupled with CN stretching
					1550.75 (PSC-IP) [37]	N–H bending vibration couples with C–N stretching vibration
					1463 (ASC Rohu) and 1454 (ASC Catla) [57]	CH_2_ bend
					1423, 1393 (ASC Rohu) and 1413, 1402 (ASC Catla) [57]	${\mathrm{COO}}^{-}$ symmetrical stretching
					1343 (ASC Rohu) and 1338 (ASC Catla) [57]	CH_2_ wagging
Amide III	1350–1220 [22]	1385	1385	The amide III bands were found at 1385 cm^−1^ for BSC and PSC.	1246.02 (ASC)^7^	NH bending
					1236 (ASC), 1236 (HWM), and 1241 (SHM) [26]	CH_2_ group wagging vibration in the glycine backbone proline side chains
					1238.94 (PSC-IP) [37]	The helical arrangement in PSC-IP.
					1240 (ASC Rohu) and 1240 (ASC Catla) [57]	NH bending coupled with CN stretching
					1083 (ASC Rohu) and 1083 (ASC Catla) [57]	C–O stretching

ASC, BSC, and PSC are collagens isolated by acetic acid, acetic acid with bromelain, and acetic acid with pepsin, respectively. ASC Rohu and ASC Catla are ASC from scales of *Labeo rohita* (Rohu) and *Gibelion catla* (Catla), respectively [57]. PSC-IP is PSC by isoelectric precipitation [37]. AAM, HWM, and SHM are collagens extracted by the acetic acid method, hot water method, and sodium hydroxide method, respectively [26].

**Table 7 marinedrugs-19-00222-t007:** Antioxidant activities of collagen samples as determined by the DPPH radical scavenging test.

Antioxidant Activity	µmol AAE/g Protein	mg AAE/g Protein
Bromelain-soluble collagen	1.03 ± 0.03	0.18 ± 0.00
Pepsin-soluble collagen	2.62 ± 0.03	0.46 ± 0.01

All values are given as mean ± standard deviation, *n* = 2. Values of antioxidant activities showed significant differences between pepsin-soluble collagen and bromelain-soluble collagen, determined by one-way ANOVA (*p* < 0.05). The straight-line equation for the antioxidant activity of ascorbic acid, used as a positive control, was: Y_1_ = −0.0193X_1_ + 0.1902, R^2^ = 0.8707; or Y_2_ = −0.0034X_2_ + 0.1902, R^2^ = 0.8707. X_1_ and X_2_ are the corrected absorbances and Y_1_ and Y_2_ are the concentrations of ascorbic acid in mg/L and µmol/L, respectively. The antioxidant activities of the collagen samples were converted into µmol AAE/g protein, or mg AAE/g protein. The IC_50_ for ascorbic acid was 3.11 ppm.

**Table 8 marinedrugs-19-00222-t008:** Antioxidant activities of collagen samples as determined by the reducing power test.

Compounds	Absorbances (700 nm)
Bromelain-soluble collagen	0.25 ± 0.00
Pepsin-soluble collagen	0.25 ± 0.01

All values are given as mean ± standard deviation, *n* = 3. Values in the same column indicate no significant differences, determined by one-way ANOVA (*p* < 0.05). The absorbance (700 nm) of ascorbic acid, based on the literature, was 3.540 [60].

**Table 9 marinedrugs-19-00222-t009:** Operating parameters for the determination of metals using ICP-OES.

Parameter	
Power (kw)	1
Plasma Argon flow rate (L/min)	15
Auxiliary Argon flow rate (L/min)	1.5
Nebulizer Argon flow rate (L/min)	0.90
Pump rate (rpm)	15
Nebulizer type	Glass concentric
Nebulizer pressure (kPa)	200
Emission line of metals (nm):	
As	188.980
Cd	214.439
Co	238.892
Cr	267.716
Cu	327.395
Pb	220.353

## Data Availability

The data presented in this study are available on request from the corresponding author.

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
