# Peer review of "Chemical and Antioxidant Characteristics of Skin-Derived Collagen Obtained by Acid-Enzymatic Hydrolysis of Bigeye Tuna (Thunnus obesus)"

_marinedrugs, 2021, doi:10.3390/md19040222_

Round 1

Reviewer 1 Report

Dear Authors, 

Please find my comments in the enclosed file. 

All the best regards

Author Response

Thank you for your review. Here, I submit again, my revised manuscript based on your advice.

Reviewer 2 Report

The article "Bigeye Tuna Skin-Derived Collagen Resulted from Enzymatic Hydrolysis with Different Proteases" describes the extraction of type I collagen from a by-product (tuna skin) and some of its properties (like antioxidant activity).

In table 2 and table 4, where histamine and some heavy metals were not detected, authors are asked to state the detection limit of the method used. Is not the same thing if their detection limit is 10mg/kg or 0.1mg/kg).

On page 3, row 102 please state the solvent for NaOH and butyl alcohol. Water seems the logical choice, but solubility of n-butanol in water cannot reach 10% for example (https://pubchem.ncbi.nlm.nih.gov/compound/butanol#section=Solubility). Also please state which type of butanol was used (normal-, 2-, iso-, tert-).

The authors should add also comparison with 2020/2021 results (the newest references are from 2019) to ensure that latest information are accounted.

Some English polishing is required (like row 263).

Therefore, I suggest publishing the paper titled "Bigeye Tuna Skin-Derived Collagen Resulted from Enzymatic Hydrolysis with Different Proteases" after minor correction in the journal Marine Drugs.

Author Response

Thank you so much for you review. Based on your advice, we have revised in our manuscript:
1. We have added the detection limit in Table 2 and Table 4.
2. The solvent used for NaOH and butyl alcohol is distilled water.
3. The butyl alcohol used is 1-butyl alcohol or n-butyl alcohol (in the materials point 3.1) (presently in line 299).
4.  We have also added 2020/2021 literature to this revised manuscript.
5. We've also revised the sentences in line 263 (presently in line 269)

Best regards,

Liza Devita, et. al.

Reviewer 3 Report

This manuscript presents a procedure to extract collagen from the lyophilized  bigeye tuna skin of using acetic acid  for the extraction.  The authors used 0.5 M acetic with and without enzymes to extract collagen.

The introduction is leading to the aim of the present work, but the authors imply that there is a benefit in using Acid extraction versus the enzyme extraction of collagen with not elaborating on this.

I believe that the authors will agree that the readers should be presented with evidence to explain and support the rationale of this work. This is crucial for evaluating a research paper.

Another issue I have with this manuscript is the similarity with a previously published paper which had very similar aim using bone as a source of collagen (Biotechnology and Bioprocess Engineering 18: 1185-1191 (2013) DOI 10.1007/s12257-013-0316-2)  I believe that the authors can use this previous paper in their literature review and discussion to support the rationale of the work and conclusions of their submitted manuscript. 

It would be nice to provide a better image of the stained gel in Figure 2 and also provide stained gels of all the extraction methods reported in this manuscript.  I wonder if the authors have any microscopy (Light or Electron microscopy) image of the collagen extracted.

The paragraph with the discussion of the results of SDS-PAGE (Lines 206-2010) needs major revision.  Please provide information (use arrows) to indicate the corresponding kDa of α1 α2 bands.  Subsequently, refer to this parameter kDa and then compare apples with apples, i.e. skin collagen with skin collagen or skin collagen with bone collage of the same species , then expand your comparison to other fish species, this will enable you to discuss differences between different species, organs and methods (the last(methods) will require much more effort to explain if you wish to insist on this line of discussion and you should include other parameters including the gel and electrophoresis conditions).

In the conclusion, please explain how you confirmed the highest collagen yield of type I collagen (see Line 459).

In view of the above, the authors can revise their manuscript to support the originality and contribution in this field to a level appropriate for a prime journal as the Marine drugs. I certainly would be happy to see a revised version of this manuscript, it is well written and the methods adequately described.

Author Response

Dear reviewers,

We would like to thank you and appreciate  your review and suggestions to improve our manuscript. We have made the changes based on your advices. The changes in the revised-manuscript are highlighted with the shading of yellow, purple, and green for reviewers 1, 2, and 3, respectively, in our revised-manuscript.

  1.  We have added the advantages of the enzymatic method in lines 56-60, and lines 72-75.
  2. We have added literature "Isolation and Characterization of Collagen from Marine Fish (Thunnus obesus)" with DOI: 10.1007 / s12257-013-0316-2, in line 139, 210-213.
  3. Figure 1. The collagen was directly photographed.  Presently, the light and electron microscopy are not available.  Due to pandemic incident various labs are not opened to the students and researchers
  4. Figure 2 was the best image from our SDS PAGE test. For other collagen samples, we did not perform SDSPAGE, since we focused on the collagens with the two highest yields, namely PSC and BSC.
  5. We apologize,  that presently we cannot add  data for other collagens because of the covid lockdown conditions that make it impossible to perform more experiments
  6. We have made a major revision at point 2.3.4 in our revised-manusrcipt (lines 198-217). Also for Figure 2, we have added red-arrows in the figure.
  7. PSC and BSC were confirmed as type I collagens from SDS PAGE, and FTIR analysis (line 468-469, 216-217, 236-237).

Best Regards

Liza devita, et.al.

(Prof. Dr. Ir. Maggy T Suhartono as corresponding author)

Reviewer 4 Report

The study presented summarizes the extraction of collagen by apllying different proteases. The overall rationale for the study was the identification and verification of alternative non-mammalian collagen sources.

Although significant improvements have obviously already been made to the present manuscript version, in my view the scientific innovation of the study is lacking. The methods of extraction have already been described, and it has also already been shown that collagen can be obtained from numerous fish species (as a mammalian alternative).

The bioactivities shown are also neither surprising nor new. Anti-oxidative properties for collagen have already been published several times. In addition, it would be interesting to show the molecular effect e.g. in keratinocytes or in a skin model. 
Since collagen extracts do not have antimicrobial properties, this data set could in principle be deleted from the manuscript altogether.
Overall, this manuscript is more of a description of material obtained using methods already described, and its biological activity has unfortunately only been described in a rudimentary way.

Author Response

We would like to thank you and appreciate  your review and suggestions to improve our manuscript. We have made various changes in our manuscript.  We have made the changes based on your advices.  

One of the protease enzymes used in this study, bromelain, has not yet been reported for extraction of  collagen from fish skin or any other sources.  Bromelain is originated from plant source, and abundantly available in tropical countries.  In this present study, it was shown potential for collagen extraction. The yield obtained was comparable to that of pepsin-collagen extraction (not significantly different by statistic analysis).

In this present study, the extraction by acid and protease was conducted at the same time, we called it as acid-enzymatic extraction, different from the previous studies which conducted a stepwise acid and enzymatic extraction.  We also discussed the FTIR characteristics of collagen more comprehensive than those in the previous studies reported by others.

We agree with you to delete the collagen activities to microbial growth, due to no activities observed.

We showed another data of antioxidant activities using reducing power method (Yildirim et al., 2001) and put the result in Table 8. The test results showed that all collagen have lower reducing power compared to that of ascorbic acid.   Nevertheless, we still feel this data is important to encourage us to go for the next step, that is adding other type of protease enzyme to produce smaller collagen peptide with higher antioxidant activities. We have made a major revision at point 2.4. (line 261-311) and point 3.6 (line 454-460) in our revised-manusrcipt for the antioxidant activities.

Best Regards

Prof Maggy T Suhartono

Reviewer 5 Report

This is a slightly modified version of marinedrugs-1129117. The study is still too preliminary. The authors must focus on biological activities of collagen-derived peptides and study them more comprehensively (e.g., antioxidant properties). According my previous review report:

“The antioxidant activity of collagen-derived peptides must be validated using at least cellular in vitro systems. Please note that antioxidant activity in cell-free assays (here, only one single assay – DPPH assay - was considered that is inadequate!!!) may not reflect antioxidant activity in biological systems. This must be addressed. The manuscript in its current form cannot be considered for publication.”

The authors ignored my comments. Instead, their provided an antimicrobial activity test but without adding experimental data (negative results were obtained). This is inadequate. Please focus on one biological activity, here antioxidant activity and provide more results on this aspect to validate the final conclusions.

Minor points:

The title is still inadequate. Please note that the title must reflect obtained results. The title in its present form is still not informative.

Author Response

We would like to thank you and appreciate  your review and suggestions to improve our manuscript. We have made various changes in our manuscript.  We have made the changes based on your advices.  

We did two different antioxidant analyses.

We show another data of antioxidant activities using reducing power method (Yildirim et al., 2001) and put the result in Table 8. The test results showed that all collagen have lower reducing power compared to that of ascorbic acid.   Nevertheless, we still feel this data is important to encourage us to go for the next step, that is adding other type of protease enzyme to produce smaller collagen peptide with higher antioxidant activities. We have made a major revision at at point 2.4. (line 261-311) and point 3.6 (line 454-460) in our revised-manusrcipt for the antioxidant activities.

We did not evaluate the antioxidant activities using biological cell method, due to the aims of application for  food.

We agree to the reviewer comment to revise the title, as follow:

Bigeye Tuna Skin-Derived Collagen Resulted from Enzymatic Hydrolysis with Different Proteases

Revised to:

Chemical and antioxidant characteristics of skin-derived collagen obtained by acid-enzymatic hydrolysis of bigeye tuna (Thunus obesus).

Best Regards

Prof Maggy T Suhartono

Round 2

Reviewer 3 Report

Dear Authors, I thank you for your effort to accomodate all the comments and suggestions. This manuscript is now improved, the introduction justifies the work presented here and your conclusions are based on the data provided. I appreciate the extra work load and the challenging conditions during the current COVID pandemic. 

Author Response

Thank you so much for your kind attention and your suggestion.

Best Regards

Liza Devita, et al.

Reviewer 4 Report

The study presented summarizes the extraction of collagen by applying different proteases. The overall rationale for the study was the identification and verification of alternative non-mammalian collagen sources.

The authors have put a lot of work into the revision and an accompanying improvement of the manuscript. It is also positive to note that the microbial studies have been taken out and some innovation (the use of bromealine) and an associated logical conclusion from the study has been inserted.

However,still the describe bioactivities shown are neither surprising nor new. Anti-oxidative properties for collagen have already been published before. In addition, it would be interesting to show the molecular effect e.g. in keratinocytes or in other human cell types. 

Author Response

Thank you and appreciate  your review and suggestions

The present antioxidant studies is regarded as the basis for our next effort in using other type of proteases to produce smaller peptides with higher antioxidant and other  types of bioactivities

We did not at present  evaluate the molecular effect in keratinocytes and other  human cell types  due to limited facilities in the in cell culture laboratories.  In addition, our aim for making the collagen is focused more on the food application

Nevertheless we have added your suggestions with statements we made in the abstract (line 24-26) and the discussion (lines 276-279 written in purple shade).

Best Regards

Prof. Dr. Ir. Maggy T Suhartono

Reviewer 5 Report

The authors have partially improved the manuscript. Cell-based evaluation of antioxidant activity and related molecular mechanisms is still lacking. This may question the biological relevance of the present pure in vitro data. The authors must highlight the limitations of the study in the abstract section and discussion section.

Author Response

Thank you and appreciate  your review

We have  added your suggestions with statements we made in the abstract (24-26) and the discussion (lines 276-279 written in purple shade).

Best Regards

Prof. Dr. Ir. Maggy T Suhartono